# How to Treat Gossip in Internet Public Carbon Emission Reduction Projects?

Zhenghong Wu [1] and Yang Sun [2,*] ⊕

1   College of Economics and Management, Civil Aviation University of China, Tianjin 300300, China
2   College of Transportation Science and Engineering, Civil Aviation University of China, Tianjin 300300, China
*   Correspondence: 2020072123@cauc.edu.cn; Tel.: +86-182-9611-2325

**Abstract:** Ant Forest is an internet public carbon emission reduction project jointly initiated by the government and enterprises and has successfully made a huge contribution to carbon reduction. As an online project, Ant Forest is more likely to receive public attention and discussion, which will undoubtedly incur a vast amount of gossip. In addition, unlike the offline acquaintance society, people need to frequently deal with heterogeneous interpersonal relationships online, which complicates the role of gossip. In this background, the impact of gossip on internet public carbon emission reduction projects and how to deal with gossip to increase public participation are important research questions. We study the above questions through public goods game. We propose three alternative coping mechanisms of gossip namely: punishment only ($P_O$), punishment with reputation compensation ($P_R$) and punishment with monetary compensation ($P_M$). The research results are shown as follows: Firstly, although the effect of gossip on advancing public participation in public carbon emission reduction projects under heterogeneous interpersonal relationships is inferior to that under homogeneous interpersonal relationship, it is undeniable that gossip also could effectively promote public to take part in internet public carbon emission reduction projects. Secondly, compared with the other two mechanisms, the mechanism $P_M$ is the most effective way to advance public participation in the internet public carbon emission reduction projects. Finally, there is optimal tolerance degree, penalty time and rebirth coefficient to maximize the promotion effect in the $P_M$. Our research demonstrates that gossip has a positive significance for internet public emission reduction projects, and we also provide policy makers with corresponding suggestions to advance public participation.

**Keywords:** carbon emission reduction; public goods game; gossip; cooperation; evolution; heterogeneity

## 1. Introduction

With the development of internet technology, internet platforms could effectively gather and allocate resources. Based on the above characteristics, more and more public carbon emission reduction projects appear on the internet, and Ant Forest is a representative. Ant Forest is a public welfare project launched by Ant Financial and China Greening Foundation to lead the public to practice low-carbon emission reduction. In the Ant Forest project, people accumulate "green energy" through actual low-carbon behaviors such as cycling, e-office and so on to apply for planting real trees in areas in need of ecological restoration to improve the environment. As of 2020, Ant Forest has successfully planted 326 million trees and reduced carbon dioxide emissions by 7.92 million tons.

In reality, we can observe that gossip could affect the public's willingness to participate in public projects. Compared with offline public projects, internet public carbon emission reduction projects similar to Ant Forests are bound to receive more attention and discussions. For instance, a lot of netizens discuss about Ant Forests on other internet forums because of the existing of opportunistic behavior of the "free-rider" in this project. In this case, people need to deal with heterogeneous relationships more frequently. It is necessary

for us to study the influence of gossip on internet public carbon emission reduction projects under the condition of heterogeneous interpersonal relationships. In addition, how to deal with gossip to promote the sustainable operation of the internet public carbon emission reduction projects is also an important problem.

The pattern of internet public emission reduction projects such as Ant Forest is similar to public goods game (PGG). So, this research is carried out on PGG. To study the above problems, we develop a public goods game model in which those who actively participate in activities of Ant Forest, such as practicing green life, collecting energy, applying for planting trees, and so on are defined as cooperators, and those users who do not contribute to carbon reduction projects but still enjoy the fruits of carbon reduction are defined as defectors.

PGG provides a powerful tool for solving the cooperation dilemma and explaining the cause of cooperative behavior among selfish individuals [1–3]. Many scholars have studied cooperation based on PGG [4]. Nowak reviewed previous studies and summarized five basic rules to promote cooperation, namely direct reciprocity, indirect reciprocity, spatial game, group selection and kinship selection. Based on the theoretical research results of Nowak, Scholars further explored the causes of cooperation and put forward many mechanisms to solve the cooperation dilemma, such as reward [5–8], punishment [9–13], emotion [14–17], imitation [18–20], social diversity [21–23], voluntary participation [24–26] and so on.

Reputation as a typical efficient mechanism to promote cooperation, has attracted the attention of many scholars [27–32]. In the classical cooperative game, scholars always follow the rational man hypothesis. However, in subsequent studies, scholars have proved that people are not completely rational, people are concerned about their face [33]. Face is one of the issues that people are concern about, and reputation is an externalized variable of face. When someone 's reputation is high, it means he saves face. When his reputation is low, it means he loses face. People will maintain their reputation at a certain level. Once the reputation is lower than this level, they will increase their reputation by adopting positive strategies to save face. Based on this perspective, scholars have conducted detailed research on reputation mainly from the aspects of effect [34], inference [35], risk [36], attitude [37], migration [38] and threshold [39] and so on. Scholars also found that gossip could promote the speed and scope of information dissemination, and a more recent stream of literature has demonstrated that gossip could amplify the role of reputation mechanism. For example, Li et al. discussed the effect of gossip on reputation [40]. Chen et al. studied the effect of different types of gossip on cooperation [41].

However, Previous studies on gossip were conducted under the following two assumptions. Assumptions 1: gossip always conforms to the facts. However, the reality is less simplistic and far more interesting. In the internet public carbon emission reduction projects, it is hard to us to judge whether gossip is consistent with the facts. For example, people often face statements that require value judgment in Ant Forest. If the research always conduct research based on the assumptions of perfect, there would be some people choosing to be a defector for being wrongly accused. Assumptions 2: the interpersonal relationship among people is homogeneous [42]. However, there is a major breakthrough in temporal and spatial constraints of communication among people in the current internet era, which makes the relationship among people tend to be heterogeneous interpersonal relationship rather than homogeneous interpersonal relationship. Based on the above two points, we should consider the effect of gossip that does not fully consistent with the facts on cooperation in heterogeneous interpersonal relationships in our study.

Firstly, we consider the conformity between gossip and facts according to the actual situation. Secondly, we divide interpersonal relationships into IT, ET and MT according to the different proportions of emotional components and instrumental components [43]. The IT is mainly composed of instrumental components, such as the relationship among ordinary internet users in the internet public social platform. Owing to the lack of emotional components among the people in IT, they do not consider favor, and censure others based

on their own subjective ideas, therefore, IT is the relationship to maximize the influence of gossip on reputation. The ET is mainly composed of emotional components, such as the relationship among the friends subscribing to each other. Owing to taking into count the emotional relationship among the people in ET, people do not debate each other, therefore, ET is the relationship in which gossip does not affect the reputation. The MT is a relationship of similar proportion of emotional components and instrumental components, such as the relationship among subscribers and bloggers of the internet platform in which people would partially consider the influence of emotional factors, therefore, in the MT, gossip has a partial influence on reputation. We assume that the influence of gossip in MT is $\tau$ times the influence of gossip in IT.

Inspired by the management of Internet social platforms, we propose three alternative solutions to the gossip, namely "punishment only ($P_O$)", "punishment with reputation compensation ($P_R$)" and "punishment with monetary compensation ($P_M$)". To be specific: $P_O$ penalizes gossipers who constantly spread information which is inconsistent with the fact but does not make any form of compensation to agents affected by the information which is inconsistent with the fact; $P_R$ penalizes gossipers who always spread information which is inconsistent with the fact and restores the reputation of agents affected by the information which is inconsistent with the fact; $P_M$ penalizes gossipers who always spread information which is inconsistent with the fact and gives appropriate monetary compensation to agents affected by the information which is inconsistent with the fact. The value of monetary compensation is *mv*, in which *m* means the times affected by the information which is inconsistent with the fact, and *v* means the value of compensation for each time affected by the information which is inconsistent with the fact.

The rest of this paper is organized as follows. Section 2 describes in detail the basic model. Section 3 presents the numerical simulation results. Section 4 gives some discussions on the obtained and draws the conclusions.

## 2. Model and Experimental Design

Let us first define the population structure of the PGG. The vertices of dynamic graph represent agents and the edge denote the pairwise partnership between agents. Initially, the coevolution of agent strategies starts from a random and homogeneous state. There are *N* nodes in the scale-free network, as depicted in Figure 1, and each of agents has $a_{vn}$ neighbors on average and an equal probability to being a cooperator who adopts the cooperative strategy or defector who adopts the defective strategy. Besides, we assume the numbers of agents and edges remain constant during the strategy of agent updating and partner-switching processes. We also assume that the proportion of gossipers, that is, those who exert an influence on the reputation of other individuals by spreading gossip in the population is a constant number *q*. Owing to the limitation of time, space, personal energy and other factors, the gossiper cannot pay attention to all people, therefore, we let the gossiper *g* spread gossip about *z* individuals in each round. We make the following assumptions about the interpersonal relationship among individuals, that is, each agent has three different types of partners and that the proportions of partners with IT, ET and MT are respectively $n_1$, $n_2$ and $n_3$, $0 \leq n_1 \leq 1$, $0 \leq n_2 \leq 1$, $0 \leq n_3 \leq 1$, $n_1 + n_2 + n_3 = 1$.

We describe the decision-making process of agent *i* who participates in the game, and influence-outputting progress of gossiper g who transmit gossip in Figure 2. In terms of the progress of influence-outputting, we first estimate whether gossip *g* is being punished, then we estimate whether he meets the conditions of being punished, and obtain the specific value of influence through the above deterministic process. In the output process of influence, we estimate the relationship between gossip *g* and the affected to get the actual reputation effect. In terms of decision-making progress, we first calculate the reputation of agent *i*, and then we obtain the strategy by estimating whether the reputation is less than the threshold. Finally, we make corresponding compensation according to whether he is affected by the information contradicting to the fact.

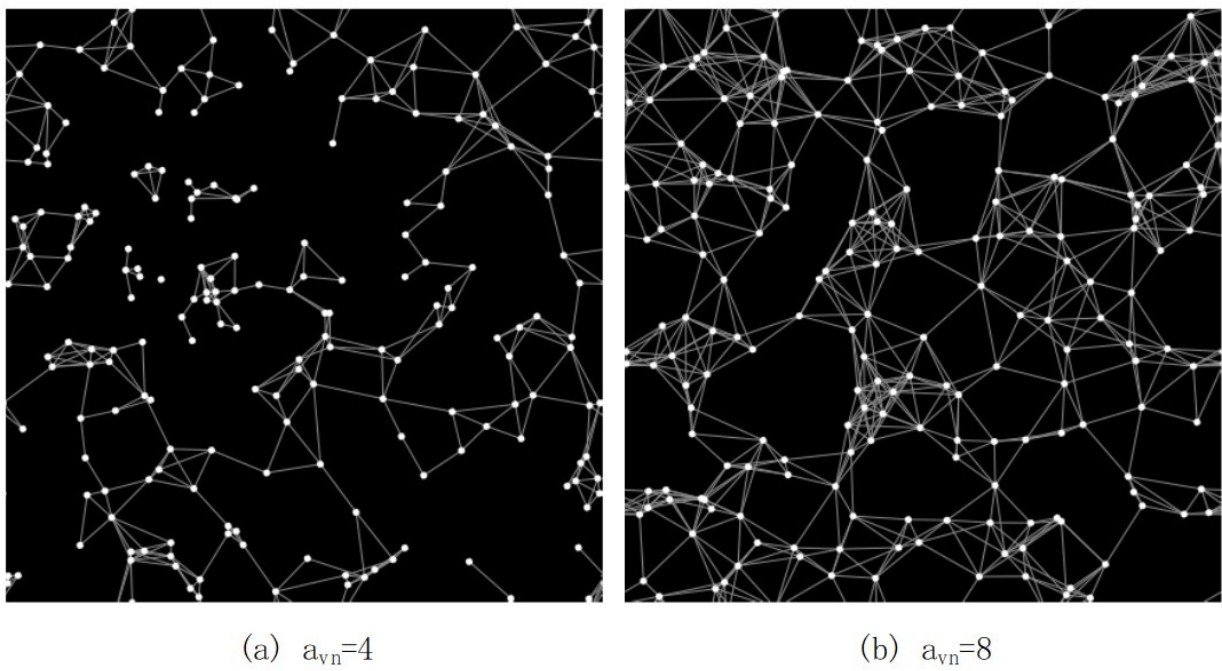

**Figure 1.** Schematic presentation of network we used in this paper. (**a**) $a_{vn}$ = 4. (**b**) $a_{vn}$ = 8. As can be seen in the figures above, some person may have many neighbors but some have few. $a_{vn}$ remain unchanged when some individuals change their group.

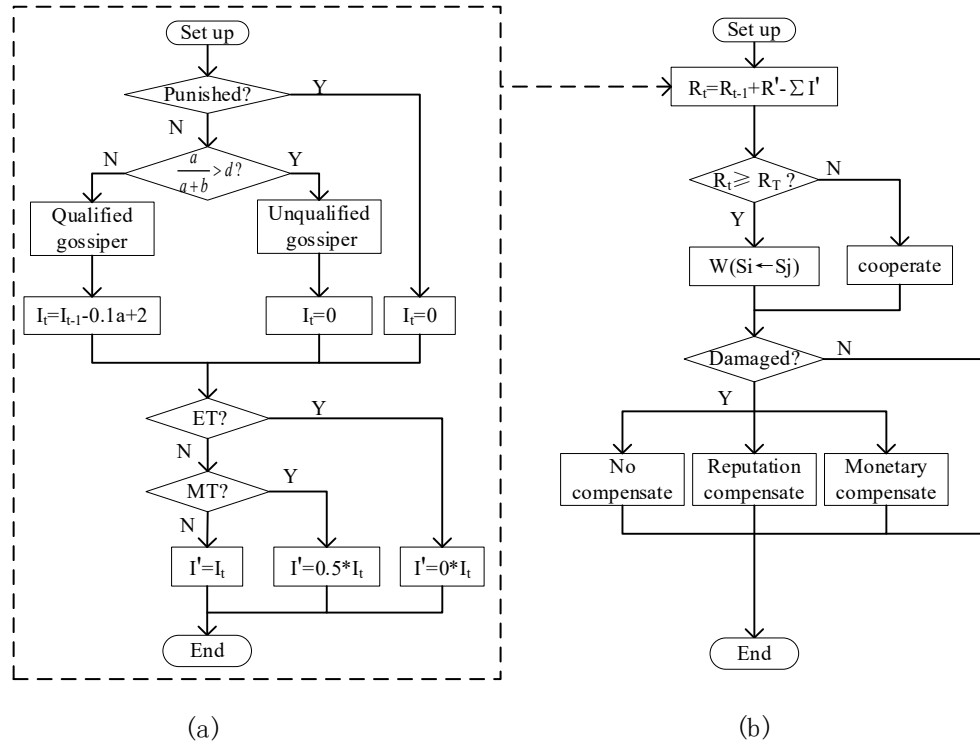

**Figure 2.** The circular process of influence-outputting *I′* of gossiper g is described in figure (**a**). The circular process of decision-making of different agent *i* is described in figure (**b**). The virtual line represents the relationship between two decision processes.

### 2.1. Payoff of Agents

In the PGG model, cooperators donate money to the common pool, and defectors do not. The total contributions in the common pool multiplied by the synergy factor *r*, and equally distributed among all agents. The payoff of agent *i* is shown in Formula (1):

$$P_i = \sum_{j \in \Omega_i} P_i^j = \sum_{j \in \Omega_i} \left( r \frac{c^j}{k_j + 1} - c_i \right) \tag{1}$$

where $P_i$ is the payoff of agent *i*, $\Omega_i$ represents the set of PGG joined by agent *i*. *j* denotes an element of $\Omega_i$. *r* (*r* > 1) denotes the synergy factor. $c^j$ represents the total amount of donations from cooperators within the group *j*, $c_i$ means the amount of contribution of agent *i*, here, we assume that the initial ci is a random variable which follows a random distribution on [0, 1], $k_j$ is the number of neighbors of agent *i* in group *j*.

### 2.2. Updating Rule of the Influence

In the dissemination of information, each gossiper has a variable influence, which refers to the ability to sway the reputation of others. Initially, the influence is a random variable which follows a random distribution on [$l_1$, $l_2$]. If $\frac{a}{a+b} > d$, the organizer regard gossiper *g* as an unqualified gossiper, otherwise they regard gossiper *g* as a qualified gossiper. Where *a* represents times of information which contradicts to the fact delivered by gossiper *g* in the last *f* rounds, *b* represents times of information which is consistent with fact delivered by gossiper *g* in the last *f* rounds, *d* means the degree of tolerance. The influence update rules for gossipers are as follows:

Gossiper *g* who is considered as an unqualified gossiper would incur penalty lasting for *h* (*h* > 1) rounds, that is, the influence of next round $I_{t+1}$ will be down to zero. The influence after penalty $I_{t+h+1}$ would restore as a value which is $\theta I_t$. $\theta I_t$ is rebirth coefficient, $I_t$ is the impact before penalty. The influence of poor gossipers is calculated as the Formula (2):

$$\begin{cases} I_{t+1} = \ldots = I_{t+h} = 0 \\ I_{t+h+1} = \theta I_t \end{cases} \tag{2}$$

When g is a qualified gossiper, the influence of g is affected by two aspects. On the one hand, it is supposed that influence decrease by *u* for delivering a piece of information which contradicts to the fact, gossiper *g* transmitted information which contradicts to the fact *a* times in recorded rounds, which reduce *g*'s influence by *ua*. On the other hand, as a reward for gossiper *g* being a qualified gossiper, the value of influence will increase *x*. The influence of qualified gossipers *g* is calculated as the Formula (3):

$$I_{t+1} = I_t - ua + x \tag{3}$$

where $I_t$ is the influence of gossiper *g* in *t* round, $I_{t+1}$ is the influence of gossiper *g* in *t* + 1 round. When *I* < 0, we set *I* = 0.

### 2.3. Updating Rule of the Reputation

Reputation is the self-perception of agents, which is an important index for agents to choose strategies. The reputation is mainly affected by gossip and strategy. When agents are talked about by gossipers, they will feel certain pressure, which leads to changes in their own reputation ratings. However, not all gossip could influence agents' self-perceptions of reputation. When gossip is from partners with ET, agent *i* would think the gossipers are joking rather than actually attacking him, so there is no actual effect on the reputation of the agent *i*. When gossip is from partners with IT, agent *i* would take gossip seriously, even doubt his own performance, and the reputation will be diminished accordingly. The value of reputation reduction is the sum of the value of impact of gossipers. When the gossip is from partners with MT, the agent *i* would partially believe in gossip, and the reputation

will partially diminished. We suppose that reputation of agent $i$ is reduced by half of the sum of impact value.

The strategy of agent $i$ is another factor affecting reputation. If agent $i$ chooses the co-operation strategy, the reputation would increase otherwise, the reputation would decrease.

At the beginning, the reputation of agent $i$ is a random variable which follows a random distribution on $[l_3, l_4]$. Reputation of agent $i$ is calculated as the Formula (4):

$$R_t = R_{t-1} + R\prime - \sum I\prime \qquad (4)$$

where $R_t$ is the current reputation. $R_{t-1}$ is the reputation of agent $i$ in the last round. $R'$ is the variation of reputation caused by the strategy of agent $i$. If agent $i$ adopts the cooperative strategy, $R'$ is a positive number, otherwise $R'$ is a negative number. $\sum I'$ is the change of agent $i$'s reputation caused by gossipers.

## 2.4. Updating Rule of Strategy

Agents' self-perception of reputation will influence their strategic choice [8]. Since reputation is related to agents' face, the value of reputation has an important impact on agents' choice of strategy. The reputation threshold $R_T$ represents the bottom line of reputation that agents could bear. Due to individual heterogeneity, the threshold of agents is different. When the value of reputation of agent $i$ is inferior to the threshold, namely $R_t < R_T$, the agent would lose face and he would choose cooperative strategies to save face. When the value of reputation of agent $i$ is not less than the threshold, that is, $R_t \geq R_T$, reputation does not affect agents' choice of strategy. Agent $i$ randomly selects a neighbor $j$ and imitates his strategy, and the specific probability of imitation is given by Fermi rule [44], as shown in Formula (5):

$$W_{(s_i \leftarrow s_j)} = \frac{1}{1 + e^{(P_i - P_j)\varphi}} \qquad (5)$$

where $s_i$ and $s_j$ represent the strategy of agent $i$ and $j$, respectively, $P_i$ and $P_j$ represent the payoff of agent $i$ and $j$, respectively, $\varphi$ is the amplitude of environmental noise. According to previous studies by scholars, $\varphi$ = 0.1 [45].

Main parameters and abbreviations in this paper are defined and explained in Table 1.

**Table 1.** The definition and description of main parameters.

| Parameters | Definition and Description |
|:---:|:---:|
| $\tau$ | The ratio of gossip impact in MT to IT |
| $m$ | The times of agent $i$ affected by the information which is opposite to the fact. |
| $v$ | The value of compensation for each time affected by the information which is opposite to the fact. |
| $n_1$ | The proportion of agent $i'$ partners with IT. |
| $n_2$ | The proportion of agent $i'$ partners with ET. |
| $n_3$ | The proportion of agent $i'$ partners with MT. |
| $a_{vn}$ | The average neighbor number of agents. |
| $q$ | The proportion of gossipers. |
| $P_i$ | The payoff of agent $i$. |
| $\Omega_i$ | The set of PGG joined by agent $i$. |
| $j$ | The arbitrary group in $\Omega_i$. |
| $c^j$ | The total contributions of group $j$. |
| $c_i$ | The total contributions of agent $i$. |
| $r$ | The synergy factor |
| $k_j$ | The number of neighbors of agent $i$ in group $j$. |
| $a$ | The times of error gossip delivered by gossiper $g$ in the last $f$ rounds. |
| $b$ | The times of correct gossip delivered by gossiper $g$ in the last $f$ rounds. |
| $d$ | The tolerance degree |
| $I$ | The impact of the gossiper |
| $R$ | The reputation of the agent |

**Table 1.** *Cont.*

| Parameters | Definition and Description |
|:---:|:---:|
| $\sum I'$ | The reputation changes caused by gossipers |
| $R'$ | The reputation changes caused by agent' decisions |
| $R_T$ | The reputation threshold |
| $W$ | The probability of agents imitating other strategies |
| $s_i$ | The strategy of agent $i$ |
| $s_j$ | The strategy of agent $j$ |
| $\varphi$ | The amplitude of environment noise |

## 3. Numerical Simulation Results and Analysis

In this section, we will show in detail the results of stable induced by gossip under different interpersonal relationships. We start from a homogeneous scale-free network, and we will give the specific parameters in the following text. Before the results, we state the initial conditions briefly. At the beginning, the population is $N = 200$, average of neighbors $a_{vn} = 4$, proportion of gossiper $q = 10\%$, time of each gossiper spread gossip $z = 4$, synergy factor $r = 2.0$, noise factor $\varphi = 0.1$. The parameters of three type partners under HT, IT, ET and MT are as follows: $n_1 = 1$, $n_2 = n_3 = 0$; $n_1 = 0.85$, $n_2 = 0.05$, $n_3 = 0.10$; $n_1 = 0.40$, $n_2 = 0.30$, $n_3 = 0.30$; $n_1 = 0.30$, $n_2 = 0.15$, $n_3 = 0.55$. Other parameters are set to be as follows: $v = 0.01$, $u = 0.1$, $x = 2$, $l_1 = 0$, $l_2 = 20$, $l_3 = 0$, $l_4 = 10$, $f = 10$, $|R| = 1$. We carried out a lot of simulations to obtain the data needed for experimental analysis, and each simulation ran at least 10,000 steps, and each parameter ran at least 20 times independently to ensure the reliability of data results. And when one of the parameters is discussed, others remain stationary. Since all simulation results can achieve comprehensive cooperation, to more accurately study the effect of gossipers on promoting cooperation, we conduct research from two perspectives: generations to equilibrium (GE) and the total contributions (TC). Here the generations to equilibrium (GE) is the speed reaching comprehensive cooperative, and the total contributions (TC) is the donation level of comprehensive cooperative. The simulation results are described in detail below.

### 3.1. Influence of Gossip on Cooperation in Different Interpersonal Relationships

We first study the influence of proportion of gossipers $q$ on cooperative evolution results under different interpersonal relationships. Figure 3 shows that all curves move downside with the increase of the proportion of gossipers. Furthermore, we can see that the black curve decreases fastest with the increase in the proportion of gossipers, and the green curve decreases slowest with the increase in the proportion of gossipers. It means that in different interpersonal relationships, there is a big difference in the change to the speed of reaching full cooperation caused by the increase of 1% of gossipers. Specifically, the speed of reaching full cooperation is most sensitive to the change of gossipers in the case of HT, while the speed of reaching full cooperation is least affected by the change of gossipers in the case ET. Through the above analysis, we can conclude that although the effect of gossip on advancing public participation in public carbon emission reduction projects under heterogeneous interpersonal relationships is inferior to that under homogeneous interpersonal relationships, it is undeniable that gossip also could play an important role in advancing public participation in public carbon emission reduction projects in the heterogeneous interpersonal relationship of the internet.

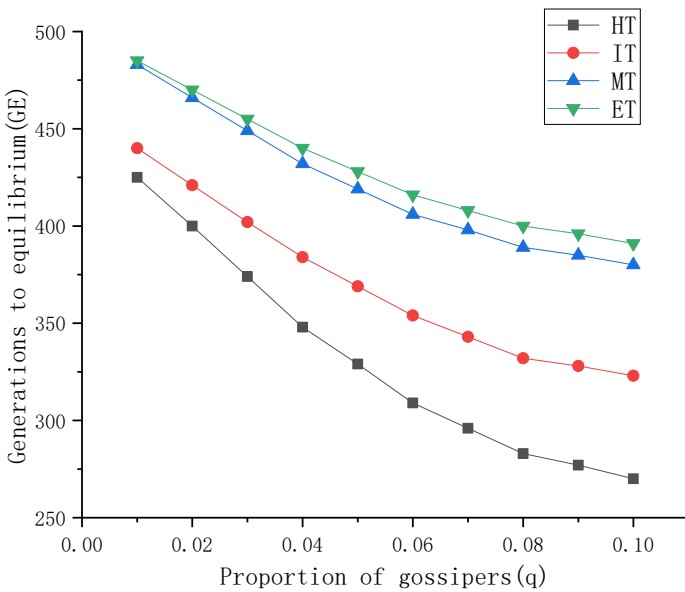

**Figure 3.** The equilibrium results of cooperation speed for different values of proportion of gossipers q varying from 0.01 to 0.10 with an interval of 0.01 under HT, IT, MT and ET.

Then, we study the influence of the conformity between gossip and facts on promoting cooperation in case of IT, MT and ET. It can be seen from Figure 4a that all curves decrease rapidly with the increase of the conformity between gossip and facts. This suggest that GE is negatively correlated with the conformity between gossip and facts in the three types of interpersonal relationships, and when gossipers spread the information which completely consistent with facts, the model achieves full cooperation fastest. In Figure 4b, we discover that the value of points is always close to 70. It demonstrates that TC is virtually unaffected by the conformity between gossip and facts. In summary, the higher the conformity between gossip and facts, the more effective it is in advancing public participation in public carbon emission reduction projects. Therefore, the internet should regulate gossip and encourage the public to investigate before speaking so as to promote the sustainable development of internet public carbon emission reduction projects.

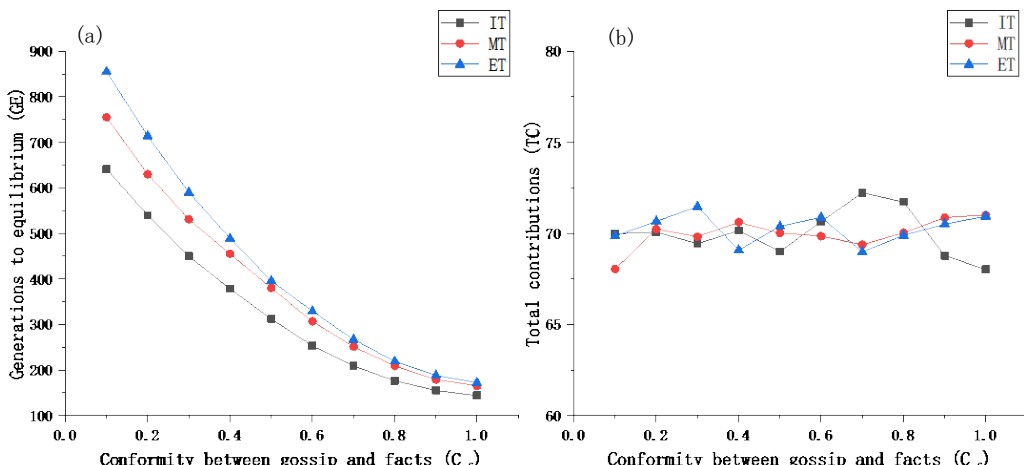

**Figure 4.** The influence of the conformity between gossip and facts $C_{gf}$ on cooperation. (**a**) The equilibrium results of cooperation speed for different values of the conformity between gossip and facts varying from 0.1 to 1.0 with an interval of 0.1. (**b**) The equilibrium results of total contributions for different values of the conformity between gossip and facts varying from 0.1 to 1.0 with an interval of 0.1.

### 3.2. The Influence of Different Mechanisms to Coping with Gossip on Cooperation

From the above research, it can be seen that the conformity between gossip and facts is, the more obvious the promotion effect on cooperation is. But the gossip in reality is not entirely consistent with reality. To deal with the above situation, we study the influence of three alternative coping mechanisms, $P_O$, $P_R$ and $P_M$ by comparing them with control group $P_N$. In the control group, we do not give agents any form of punishment and compensation. In subsequent studies, we suppose that the conformity between gossip and facts is 0.5. From Figure 5a–c, we can see that in three types of interpersonal relationships, the height of yellow bar and the red bar always close together, the blue bar is the shortest one and the green bar is the tallest one. It means that from the speed perspective, $P_M$ is the best coping mechanism, $P_R$ is the worst coping mechanism and $P_O$ and $P_N$ are intermediate-level coping mechanisms. Similarly, in the line chart, $P_M$ is the tallest one and the remaining points are at a similar height. From the contribution perspective, $P_M$ is also the best coping mechanism. In summary, the coping mechanism $P_M$ can effectively promote cooperation. In order to explore the reason why mechanism $P_M$ promotes cooperation in more detail, we further draw Figure 5d, which illustrates the composition of contribution in the case of IT, MT and ET. From Figure 5d, we discover that the blue bars are tallest, and the portion of oblique shadow which represents agents whose donation level between 0.7–1 is more than 70% in IT, MT and ET. The line chart shows that the proportion of compensated agents is less than 40%. There is an obvious gap between high-level cooperators and compensated agents. We speculate that there are two reasons why $P_M$ improves the level of donation. On the one hand, part of the compensated agents take out a portion of the monetary compensation to the common pool to increase their contributions. On the other hand, the compensated agents play a good demonstration effect, and some of other agents chose to donate more money because they are influenced by compensated agents. Since $P_M$ has the best effect on promoting cooperation than other mechanisms, in the following study, only mechanism $P_M$ is considered, and the other two mechanisms are not.

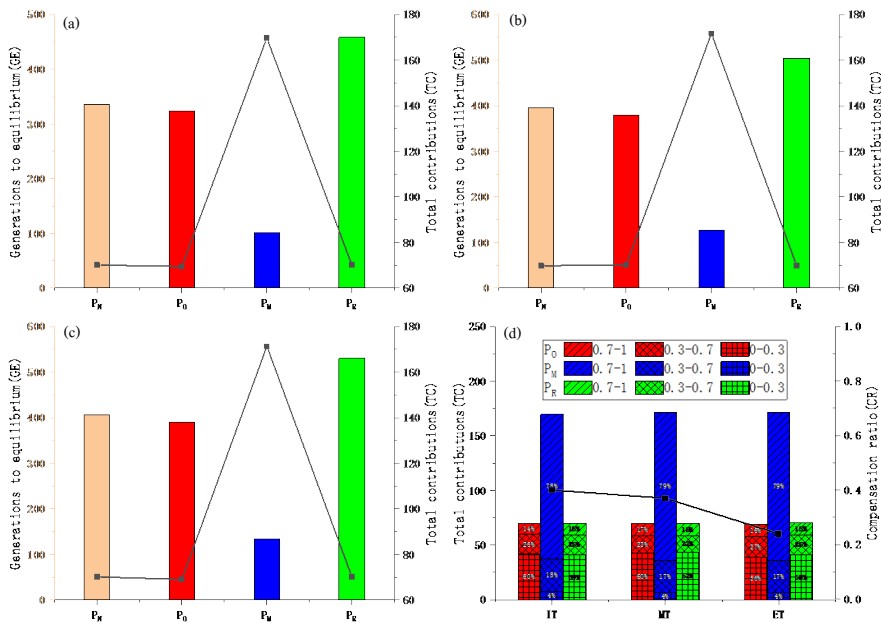

**Figure 5.** The influence of $P_O$, $P_R$ and $P_M$ on cooperation in different interpersonal relationships. Orange represents the influence of $P_N$, red represents the influence of $P_O$, blue represents the influence of $P_M$, green represents the influence of $P_R$. (**a**) The evolution results of influence of $P_O$, $P_R$ and $P_M$ on cooperation in the case of IT. (**b**) The evolution results of influence of $P_O$, $P_R$ and $P_M$ on cooperation under MT. (**c**) The evolution results of influence of $P_O$, $P_R$ and $P_M$ on cooperation under ET. (**d**) Composition of donation in the case of IT, MT and ET. The main parameters are as follows: $d = 0.4$, $h = 5$, $\theta = 0.5$.

### 3.3. Three Specific Penalty Details of $P_M$

We further examine the detail of $P_M$. We first examine the influence of degree of tolerance $d$ on cooperation, and the results are shown in Figure 6. From Figure 6a, we find that all the curves are similar to the smile curve, and there is an optimal interval of degree of tolerance around 0.3. In terms of speed to achieve comprehensive cooperation, the optimal value of degree of tolerance is 0.3. From Figure 6b, all curves are twisted together around 170. It means the influence of changes in degree of tolerance on total contributions could be ignored. Figure 6c shows that curves present a unified increasing trend, which means when the degree of tolerance is 0.1, the compensation of the model is at the lowest level. By comparing the above figures, especially Figure 6a,c, we find that there is a contradiction between the speed of comprehensive cooperation and compensation cost, that is, when the speed is optimal, the cost is not the lowest, and vice versa. In order to better play the role of the coping mechanism, we introduce a new index $EC = lg\ (GE + TC)$ to select the optimal degree of tolerance. EC represents a comprehensive cost, when EC reaches the minimum value, the model could quickly achieve comprehensive cooperation with a small compensation cost. In Figure 6d, all curves also decrease first and then increase, and the optimal value is 0.3. Based on the above conclusions and analysis, when the degree of tolerance is maintained at a relatively low level, it is most conducive to the internet public carbon emission reduction projects.

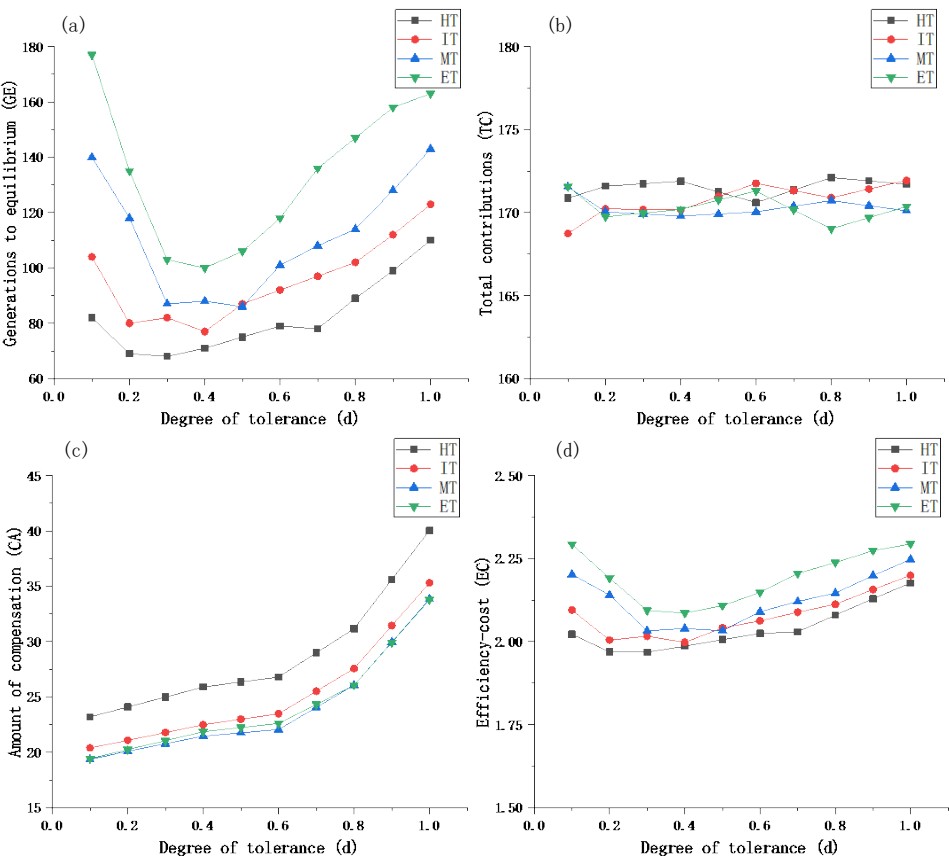

**Figure 6.** The influence of tolerance degree d on cooperation. (**a**) The equilibrium results of speed of achieving comprehensive cooperation for different values of tolerance degree d varying from 0.1 to 1.0 with an interval of 0.1. (**b**) The equilibrium results of total contributions for different values of tolerance degree d varying from 0.1 to 1.0 with an interval of 0.1. (**c**) The equilibrium results of compensation for different values of tolerance degree d varying from 0.1 to 1.0 with an interval of 0.1. (**d**) The equilibrium results of efficiency-cost for different values of tolerance degree d varying from 0.1 to 1.0 with an interval of 0.1. The main parameters are as follows: h = 5, θ = 0.5.

We continue to study the influence of penalty time *h* on cooperation. From Figure 7a, all curves show an increasing trend of S-shaped curve, and when the penalty time is less than 35, the curve increases with the increase of punishment time, when the punishment time is greater than 35, the curve is almost no longer affected by the penalty time. It is demonstrated that in terms of speed to achieve comprehensive cooperation, the optimal penalty time is 1. All curves in Figure 7b maintain at an approximate height, which indicates that the penalty time almost does not affect the total contributions. In Figure 7c, the curve shows a decreasing trend, the same as the effect on velocity is that when the punishment time is greater than 35, the curve no longer decreases with the increase of penalty time. Therefore, when the penalty time is 35, the compensation cost the model is at the floor level. It can be seen from Figure 7a,c that the impact of penalty time on speed and compensation is still contradictory. We continue to utilize the index EC, and the results are shown in Figure 7d. All curves increase as the penalty time increases, and when the penalty time is 1, EC gets minimum value. Based on the above analysis, short-term punitive measures could effectively promote the healthy development of internet carbon emission reduction projects.

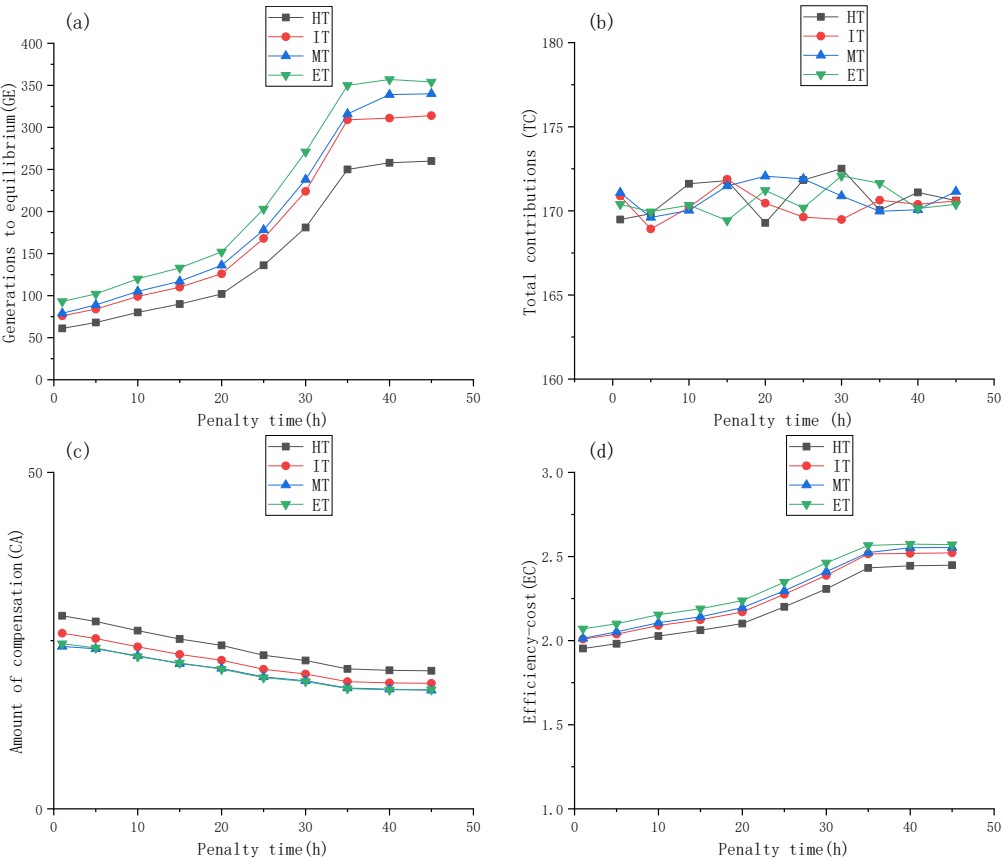

**Figure 7.** The influence of penalty time h on cooperation. (**a**) The equilibrium results of cooperation speed for different values of penalty time h varying from 1 to 45. (**b**) The equilibrium results of total contributions for different values of penalty time h varying from 1 to 45. (**c**) The equilibrium results of compensation for different values of penalty time h varying from 1 to 45. (**d**) The equilibrium results of efficiency-cost for different values of penalty time h varying from 1 to 45. The main parameters are as follows: d = 0.4, θ = 0.5.

After the punishment, we study the influence of the rebirth coefficient *θ*. From Figure 8a, when the rebirth coefficient is less than 0.9, all curves move downside with the increase of rebirth coefficient. When the rebirth coefficient is greater than 0.9, all curves run upward with the increase of rebirth coefficient. In terms of speed of cooperation,

the optimal value of rebirth coefficient is 0.9. From Figure 8b, all curves are at the same height, in other words, rebirth coefficient has little effect on total contributions. Figure 8c shows that with the increase of rebirth coefficient, the curve shows an increasing trend. Therefore, when the rebirth coefficient is 0.1, the compensation cost of the model is at the floor level. We continue to use the index EC, and the results are shown in Figure 8d. All curves decrease as the rebirth coefficient increases, when the rebirth coefficient is greater than 0.9, all the curves show a slight increasing trend. When the rebirth coefficient is 0.9, the effect of promoting cooperation is most obvious. Based on the above analysis, the internet should take a tolerant attitude towards the punished people and encourage them to return to the internet carbon emission reduction projects. But tolerant without a bottom line is undesirable, there must be some punishment as a warning.

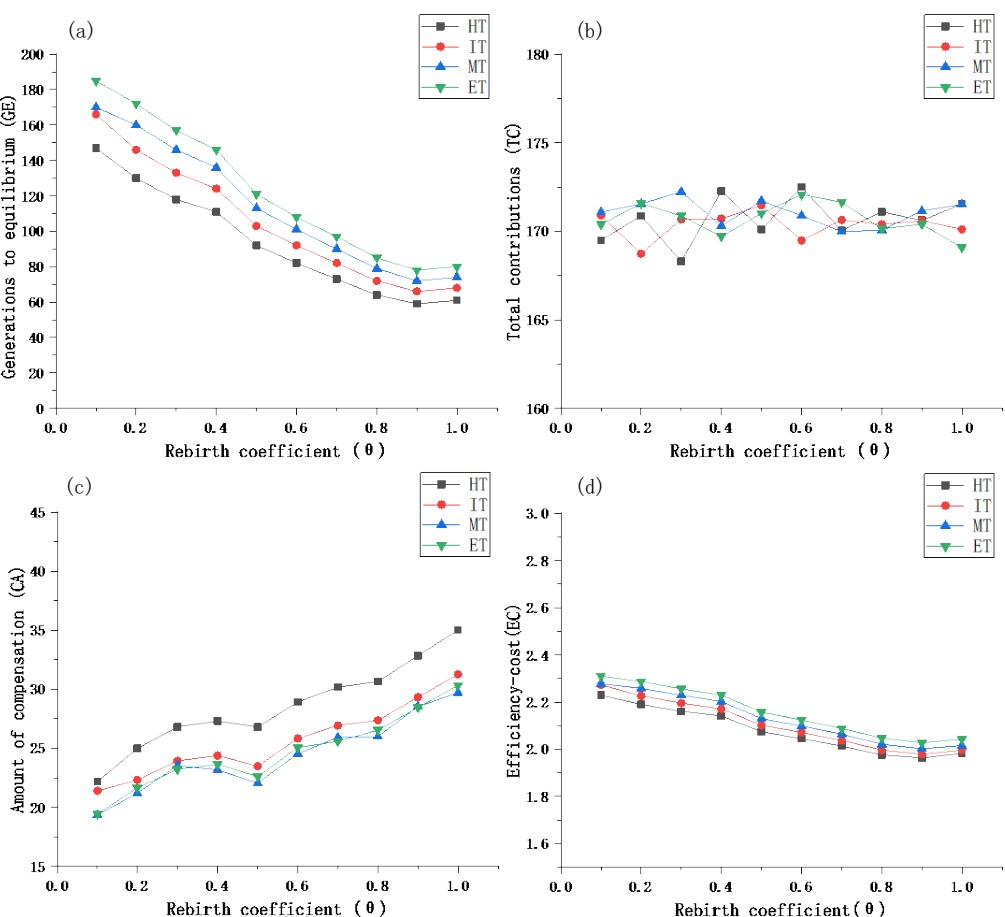

**Figure 8.** The influence of rebirth coefficient θ on cooperation. (**a**) The equilibrium results of cooperation speed for different values of rebirth coefficient θ varying from 0.1 to 1.0 with an interval of 0.1. (**b**) The equilibrium results of total contributions for different values of rebirth coefficient θ varying from 0.1 to 1.0 with an interval of 0.1. (**c**) The equilibrium results of compensation for different values of rebirth coefficient θ varying from 0.1 to 1.0 with an interval of 0.1. (**d**) The equilibrium results of efficiency-cost for different values of rebirth coefficient θ varying from 0.1 to 1.0 with an interval of 0.1. The main parameters are as follows: d = 0.4, h = 5.

### 3.4. Robustness of the Model

Finally, we study the influences of the synergy factor $r$ and the average number of neighbors $a_{vn}$ to study the robustness. In Figure 9a, with the increase of the synergy factor, the color of the picture progressively changes from red to blue. When the synergy factor is greater than 2, the color no longer changes. In addition, the color is not affected by the change of the average number of neighbors. Similar to Figure 9b, with the increase of

the synergy factor, the color of Figure 9b gradually changes from blue to red. When the synergy factor is greater than 2, the color no longer changes too. When the average number of neighbors increases steadily, the color does not change. It means that the equilibrium results are robust against the parameter $r$ and $a_{vn}$ in our model.

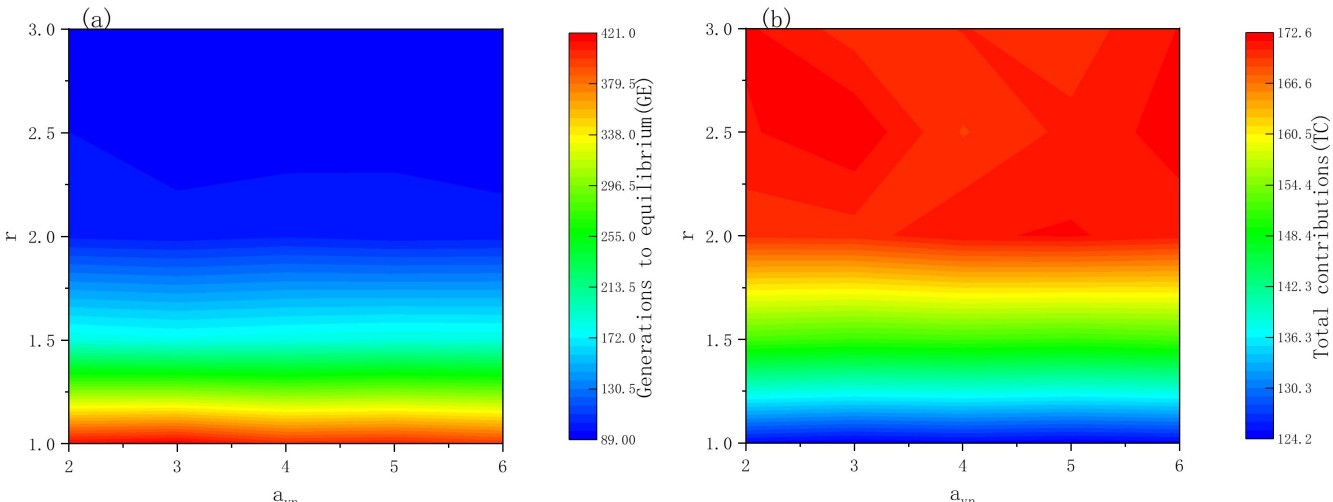

**Figure 9.** The influence of synergy factor r and average number of neighbors $a_{vn}$ on cooperation. (**a**) Heat-maps of cooperation speed at equilibrium along 2D plain based on average number of neighbors $a_{vn}$ and synergy factor r. All kinds of colors represent various speed of achieving comprehensive cooperation under the joint action of different values of $a_{vn}$ and r. (**b**) Heat-maps of total contributions at equilibrium along 2D plain based on average number of neighbors $a_{vn}$ and synergy factor r. All kinds of colors represent various total contributions under the joint action of different values of $a_{vn}$ and r. The *X*-axis is $a_{vn}$ (from 2 to 6) and the *Y*-axis is r (from 1 to 3). The main parameters are as follows: d = 0.4, θ = 0.5, h = 5.

## 4. Discussion

Different from previous scholars who study carbon emission reduction projects from the perspective of government management [46,47], we study internet carbon emission reduction projects from the perspective of public participation. We find that the essence of internet public carbon emission reduction projects is similar to the public goods game, so we use public goods game model to study the internet carbon emission reduction projects.

As one of the factors that effectively promote public cooperation, gossip has attracted the attention of many scholars. However, in the study of gossip, previous scholars only consider the homogeneous interpersonal relationships and ignored the heterogeneity of interpersonal relationships [41,42]. Obviously, it is not consistent with the reality. In order to get more realistic results, we studied the role of gossip in heterogeneous interpersonal relationships.

Previous scholars have made some achievements in the study of punishment. However, scholars only pay attention to the process of punishment itself, but ignore the matters after punishment [10]. In this article, we consider the whole process of punishment. We not only focus on the process of punishment, but also compare three different compensation measures. In addition, we also study matters after the end of punishment.

Through simulation experiments, the research results we obtained are also different from previous scholars. Firstly, we prove that gossip plays an important role in promoting cooperation in heterogeneous interpersonal relationships, which is similar to the previous scholars' research. However, most of the previous studies were conducted under the assumption of homogeneous interpersonal relationships, and the results were higher than the effects in reality. Secondly, previous studies defaulted to the accuracy of gossip, but we verify that the degree of consistency between gossip and facts has a significant impact on the cooperation of public utilities, which is not considered by previous scholars.

Thirdly, we innovatively compare the impact of three coping mechanisms on cooperation, and conclude that the punishment with monetary compensation is the best mechanism. This is also the content that scholars paid less attention to before. Finally, we propose the governance measure after punishment, that is, managers should actively encourage punished people to participate in public comments.

## 5. Conclusions

With the development of the internet, the government or enterprises launched public carbon emission reduction projects on the internet and have made some achievements. Compared with offline projects, internet projects are more susceptible to public attention and discussion, and are also more susceptible to gossip. In order to promote the sustainable development of the internet public carbon emission reduction projects, we study the influence of gossip on advancing public participation in these projects and how should we actively deal with gossip. We made the following assumptions in the study. Firstly, the internet public carbon emission reduction project is a public goods game. Secondly, people are bounded rational rather than completely rational individuals. We draw the following conclusions through the research.

(i) Gossip could also play an important role in advancing public participation in internet public carbon emission reduction projects under the heterogeneous interpersonal relationship and encouraging public to investigate before spreading information is conducive to improve the effect of gossip. Based on the role of gossip, we should pay more attention to the influence of gossip in the real world and improve the quality of gossip. (ii) Among the three alternative mechanisms we propose, $P_M$ is most conducive to the internet public carbon emission reduction projects. Therefore, in real life, we should not only pay attention to the people who spread false information, but also pay attention to the people who are affected by the information. (iii) In addition, we find that when using mechanism $P_M$, there is a desirable optimal interval for the degree of tolerance, penalty time and rebirth coefficient to maximize the public participation, that is, to adopt a low level of degree of tolerance, transient punishment time and high rebirth coefficient is conducive to the development of these projects. The conclusion reminds us that in real life, we should take an appropriate attitude towards people who spread false information.

Based on the above conclusions, in order to advance public participation in internet public carbon emission reduction projects, we put forward the following suggestions for internet managers. (i) The increase in the proportion and quality of gossip is helpful to increase public participation in internet public carbon emission reduction projects, so the manager should encourage public to actively comment on public carbon emission reduction projects and spread fact-based information. (ii) From the above conclusions, the mechanism $P_M$ promotes the public participation in internet public carbon emission reduction projects most effectively, so the manager could punish those who continually spread information which contradicts the fact and compensate those who are affected by the above information with money. (iii) According to the above results of degree of tolerance, penalty time and rebirth coefficient, the manager should take harsh but short-term measures to punish those who continually spread information which contradicts to the fact, and also readmit them with a tolerance attitude.

In summary, our research conclusions put forward some governance suggestions for internet public carbon emission reduction projects, and we hope these conclusions could promote development of these projects. Our research does not consider the cost of identifying gossip on the internet, and the following research can be carried out on this issue.

**Author Contributions:** Conceptualization, Z.W. and Y.S.; methodology, Y.S.; software, Y.S.; validation, Z.W. and Y.S.; formal analysis, Y.S.; data curation, Y.S.; writing—original draft preparation, Y.S.; writing—review and editing, Z.W. and Y.S.; visualization, Y.S.; supervision, Z.W.; funding acquisition, Z.W. All authors have read and agreed to the published version of the manuscript.

**Funding:** This research was funded by the National Natural Science Foundation of China Grant No. 72104237 and the Research Start-up Fund of Civil Aviation University of China Grant No. 53 2020KYQD08.

**Institutional Review Board Statement:** Not applicable.

**Informed Consent Statement:** Not applicable.

**Data Availability Statement:** The data that support the findings of this study are available from the corresponding author, Yang Sun (author initials), upon reasonable request.

**Conflicts of Interest:** The authors declare no conflict of interest.

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
