# Peer review of "How to Treat Gossip in Internet Public Carbon Emission Reduction Projects?"

_sustainability, doi:10.3390/su141912809_

Round 1

Reviewer 1 Report

This paper presents some findings from a model and experiment in development of the internet public carbon emission reduction projects. The authors study the influence of gossip on advancing public participation of these projects; the paper discusses how one should actively deal with gossip. This is important for any public good set-up.

The analysis and thereby the conclusions presented here confirm that gossip does play an important role in public carbon emission reduction projects. The paper also discusses the effect of the degree of tolerance, punishment time with money and reputation effect.

My main comment is that the authors should have presented these effects in terms of any public good set-up and perhaps compare the current model with known results in the literature (on public good games).

The authors also must do an overall good editorial check. There are plenty of “mistakes” both in presentation and in formatting. This is really not acceptable in this age!

To sum up, I do think this paper has discussed and presented a nice “experiment” study and I am thus happy to recommend a suitably revised version be accepted for publication, however, the revision should be written properly, with good care.

Reviewer 2 Report

Dear Authors

Your article is very interesting, well-written and very clear, I am grateful for the opportunity to read it. I think that idea and the subject of the research are very interesting, and the results of the research give a lot of new information and possibilities of further analysis.

Reading the text, I found only 3 elements (abstract, discussion and conclusions) those I think would improve your article.

First issue: the abstract. The abstract should contain background information that helps a reader to know what is in the paper, what is the purpose of the paper, what methods were used, and what are the general conclusions (partially of course it does). Because any abstract should clearly define the purpose of the research and further analysis, the research method, the subject of research, and the results. A good abstract structure is needed by the reader, but it is also important for you as authors - readers often use the abstract review method to search for content that interests them. Your abstract is not well structured, and this may be a barrier to popularizing your article. I miss the research method and answer to the question about the context of the research.

Main part: Both the introduction and the literature review are well prepared. The research is presented very well - both the formal and descriptive part as well as the presentation of the results. I believe you did your job very well. However, I have the impression that the discussion of the research results both in the context of reference to the research results of other authors and to other opinions and concepts, as well as in terms of the summary of the entire article, are clearly weaker than the rest.

In my opinion, much more important than the research itself are the implications and conclusions it brings both for science and in the context of how the research results are presented in general. So, first you should work on the chapter "Discussions". To increase the significance of the results, the discussion part should embrace the differences and similarities among your findings and those of other scholars. There is clearly no analysis of other studies. Just as there is no reference to the real situation. There is no reference to the broader perspective and thus no real discussion. Consequently, all this reduces the article to the research part and causes the reader to evaluate the quality of the research, not its purpose or conclusions.

Second, “Conclusions” this is the most important chapter, there you can find the results, your view your opinion and links between your results and real world. So, the conclusion section should be a summary of article’s aim, methods, and findings. It is not here. In my opinion conclusions are insufficient - this chapter should be extended. There is neither no reference to your assumptions. At this point, you should show references to your research and all formal aspects of your article. At the begging (introduction) and at the end (Conclusion) you should include a description of the research questions and/or research hypotheses. You should develop and explain your goals. It is necessary to change the convention from the presentation of research to the presentation of results and conclusions.

Summarizing.

I really like your article and appreciate your work. It is interesting topic, and the conclusions could open the way for further research. In my opinion You should make some changes in Discussion and Conclusions. But my general opinion and my assessment of your research and whole article is more than positive.

Good luck! 
